# Innovation indicators based on firm websites —Which website characteristics predict firm-level innovation activity?

**Janna Axenbeck**[1,2]* , **Patrick Breithaupt**[1,2]

**1** Department of Digital Economy, ZEW – Leibniz Centre for European Economic Research, Mannheim, Germany, **2** Justus-Liebig-University Giessen, Faculty of Economics, Gießen, Germany

☯ These authors contributed equally to this work.
* janna.axenbeck@zew.de

**Data Availability Statement:** The EU-GDPR as well as the German Data Protection Act (BDSG) only allow the employed firm-level data to be accessed in research data centers. Contact details for the research data center that administers the firm-level

## Abstract

Web-based innovation indicators may provide new insights into firm-level innovation activities. However, little is known yet about the accuracy and relevance of web-based information for measuring innovation. In this study, we use data on 4,487 firms from the Mannheim Innovation Panel (MIP) 2019, the German contribution to the European Community Innovation Survey (CIS), to analyze which website characteristics perform as predictors of innovation activity at the firm level. Website characteristics are measured by several data mining methods and are used as features in different Random Forest classification models that are compared against each other. Our results show that the most relevant website characteristics are textual content, the use of English language, the number of subpages and the amount of characters on a website. In our main analysis, models using all website characteristics jointly yield AUC values of up to 0.75 and increase accuracy scores by up to 18 percentage points compared to a baseline prediction based on the sample mean. Moreover, predictions with website characteristics significantly differ from baseline predictions according to a McNemar test. Results also indicate a better performance for the prediction of product innovators and firms with innovation expenditures than for the prediction of process innovators.

## 1 Introduction

Innovation, defined as the implementation of either new or significantly improved products or processes as well as combinations thereof [1], brings vast benefits to consumers and businesses. Moreover, technological progress is considered as a main driver of economic growth [2]. It is, therefore, a matter of public interest to analyze and understand innovation dynamics as it is conducted in several studies (e.g., [3–9]).

A prerequisite for the analysis of innovation-related questions is to correctly measure firm-level innovation activities. However, it should be noted that no universally accepted measurement approach exists. For example, firm-level innovation indicators are traditionally

data, the ZEW Research Data Centre (ZEW-FDZ), can be retrieved from https://www.zew.de/en/research-at-zew/zew-research-data-centre-zew-fdz. The website data can also not be published as imposed by the German Copyright Act (§60d III p. 2). The data is stored in ZEW's Dark Archive. For scientific review and verification it is possible to request access at https://www.zew.de/en/das-zew/serviceeinheiten/zentrale-dienstleistungen/bibliothek/.

**Funding:** The German Federal Ministry of Education and Research provided funding for the research project (TOBI - Text Data Based Output Indicators as Base of a New Innovation Metric; funding ID: 16IFI001, Dr. Georg Licht) https://www.bmbf.de/en/index.html. The funders had no role in study design, data collection and analysis, decision to publish, or preparation of the manuscript.

**Competing interests:** The authors have declared that no competing interests exist.

constructed with data from large-scale questionnaire-based surveys like the biennial European CIS or the annual MIP (see [10, 11]), which is also the German contribution to the CIS. However, these innovation indicators suffer from some major drawbacks (i.e., [12–14]). For instance, the MIP annually surveys around 18,000 firms. This only corresponds to a fractional share of the total stock of German firms and therefore lacks regional granularity and coverage. In addition to this, questionnaire-based surveys—especially on a large scale—have the added disadvantages of being costly and a lack of timeliness. Also, most surveys require firm participation and as a consequence, surveys such as the MIP suffer from low response rates [12]. Besides, firm-level innovation can also be studied by patent or publication analysis. However, respective indicators only cover technological progress for which legal protection is sought [15, 16] and not every innovation can be patented. For example, due to the German regulatory framework it is quite difficult to patent software, i.e., digital innovations.

Issues, however, could be solved by adding web-based information: Advances in computing power, methods for statistical learning as well as natural language processing tools enable, e.g., researchers to extract website information on a large scale. This makes it technically possible to complement traditional innovation indicators with information from scraped firm websites. Nowadays, almost every firm has an online presence. Firm websites can entail information about new products, key personnel decisions, firm strategies, and relationships with other firms [17]. Those pieces of information might be directly or indirectly related to a firm's innovation status. By using this information, it is possible to conduct an automatic, timely and comprehensive analysis of firm-level innovation activities, as measurements can be carried out faster and in shorter intervals in comparison to traditional indicators.

The contribution of this paper to the question whether web-based innovation indicators are feasible is threefold. First, we analyze to what extent firm websites improve predictions of firm-level innovation activity. Second, we assess which characteristics of a website relate most to a firm's innovation status. Third, we examine which characteristics are appropriate for predicting different forms of innovation activity. We test the latter by additionally comparing the predictive power of different innovation indicators related either to product innovations, process innovations or innovation expenditures. We assume differences between indicators, for example, because firms with process innovations may have a smaller incentive to announce respective innovation activity. This may be due to the fact that new processes are less relevant for most website visitors.

For our analysis, data on 4,487 German firms from the MIP 2019 is used. We extract their websites' text and hyperlink structure by applying the ARGUS web-scraper [13]. Several methods including topic modelling and other natural language processing tools are applied to generate features that potentially relate to the firm-level innovation status. Furthermore, we extract information related to a website's technical maturity such as how fast it is responding and whether a version for mobile end user devices is available. After extracting and calculating a wide variety of features, we divide them into three different feature sets: I) text-based features including, e.g., words, document-topic probabilities derived from a topic modelling algorithm, and the share of English language, II) meta information features including, e.g., website size related features, availability of a mobile version and loading time, and III) network features including, e.g., hyperlinks to social networks as well as incoming and outgoing hyperlinks. Based on these three feature groups, we analyze which website characteristics best predict a firm's innovation status reported in the MIP 2019 by using a Random Forest classifier.

Our results show that predictions based on website characteristics can perform significantly better than a random prediction based on the sample mean. Consequently, firm websites entail information that relate to firm-level innovation activity. In addition, our website characteristics better predict firms with product innovations and innovation expenditures than with

process innovations. Moreover, text features make the biggest contribution to our prediction performance.

Evaluating the predictive power of single variables across feature sets by means of the mean decrease in impurity (MDI), the language of a website and website size measured by the number of subpages as well as the total amount of characters are always relevant in the models with the highest predictive power for all considered innovation indicators. Moreover, there are characteristics that are highly important only for specific indicators, e.g., the verb "to develop" is more important for innovation expenditures and product innovators than for process innovators.

The remainder of this paper is structured as followed: Previous literature is reviewed in Section 2. In Section 3, we present our data and in Section 4 the descriptive statistics. Section 5 describes the methodology and Section 6 shows the results, which are discussed in Section 7. This paper concludes in Section 8.

## 2 Literature review

The usage of text data to generate innovation-related indicators has been tested in previous studies. For example, [18] show that the significance, i.e., relevance, of a patent is higher when its textual content is very distinct to previous patents but similar to subsequent ones. [19] generate innovation-related topics from 170,000 technology news articles using a Paragraph Vector Topic Model. They analyze the diffusion of the identified topics within the text corpus. Their results suggest that technology trends can be assessed by measuring the importance of topics over time. Using PATSTAT data, [20] show that context similarity of technological codes relates to innovative events. The likelihood that new combinations of technological codes appear in one patent can be predicted by their context similarity in patents where they have been used before.

Remarkable work is also conducted by [21]. In this study, a Latent Dirichlet Allocation (LDA) model is fitted with analyst reports of firms included in the S&P 500 index. The LDA topic that has the lowest Kullback-Leibler divergence to the wording of a mainstream economic textbook on innovation is chosen as innovation indicator. The authors show that firms have patents with greater impact (i.e., more citations per patent) if the innovation topic has a larger share in their analyst report. However, analyst (or also annual) reports are not available for every firm and smaller firms are particularly underrepresented. In contrast, firm websites are available for a large share of small and medium-sized firms.

Furthermore, previous literature shows that information produced online can be used to construct frequent real-time estimates [22]. Famous 'now-casting' examples that utilize web-based information are [23], who use Google search queries to accurately predict influenza activity in the United States. [24] claim that search engine query indices are also often correlated with economic activities and enable to generate frequent indicators. They show that forecasts concerning, for example, automobile sales and unemployment can be significantly improved by including search term indices in prediction models. Not only information from online searches but also firm website information can be used to generate economic indicators: As they provide detailed information about the firm as well as its products, they appear to be suitable for measuring firm-level innovation activities [17]. [13] summarize previous studies that analyze the possibility of firm website-based innovation indicators (e.g., [17, 25–29]). Most studies solely focus on the hyperlink structure of websites or only conduct a simple keyword search and are limited to small amounts of firms from a particular economic sector.

Firstly applying advances in statistical learning, [30] attempt to predict innovation at the firm level using textual information on websites and novel machine learning tools. They use a

questionnaire-based firm-level product innovation indicator (innovative/ non-innovative) from the MIP years 2015-2017 as a target variable to train an artificial neural network classification model on website texts. The authors only consider stable product innovators in their main analysis. Firms that switch between innovation statuses, which is highly relevant in the field of innovation economics, are only observed in a secondary analysis. The average F1-score for the respective prediction is 0.68%. Additionally, [14] fit several machine learning models to develop a firm website-based innovation indicator, with their annotated data set being limited to 500 firms. One important characteristic of their work is the individual analysis of websites' subpages instead of predicting the innovation status of an entire website, i.e., firm. Additionally, their subpages are manually labelled as either innovation or non-innovation-related messages instead of using survey or patent data as target variables. The best performance is achieved with an artificial neural network. Even though the predictive performance is very high, the authors cannot show a credible external validity of their indicator.

Furthermore, another issue of both approaches is that neural networks do not reveal any decision rule that can be easily interpreted by humans, which is why they are often called black box models. It should also be noted that both studies only consider text. Nonetheless, previous results show that there must be distinct website characteristics that relate to a firm's innovation status, but the particular website characteristics are not identified yet.

[31] analyze whether firm's expenditures on innovation can be predicted by means of administrative records and balance sheet data. Using a Random Forest regression approach, the authors identified firm size, sectoral affiliation and investment in intangible assets as the most important predictors. Random Forests usually provide better predictive performance than linear methods while retaining the interpretability of feature relevance.

By applying a Random Forest approach to a large scale firm-level data set, we are able to analyze which website characteristics are linked to firms' innovation activity and are highly predictive. One further contribution of our paper is to address shortcomings of previous literature, as it provides new and detailed insights on the question whether firm websites entail measurable information on firm-level innovation activities.

## 3 Data

Based on the Oslo Manual, in our data set an innovation is defined as "a new or improved product or process (or combination thereof) that differs significantly from the unit's previous products or processes and that has been made available to potential users (product) or brought into use by the unit (process)" [1, p. 20]. Furthermore, we consider all expenditures spent for innovation purposes as innovation expenditures and summarize firm-level product or process innovation as well as innovation expenditures as innovation activity.

We use data from the MIP 2019 to classify firms as either innovative or non-innovative [32]. The MIP is an annual survey conducted by the ZEW—Leibniz Centre for European Economic Research. The survey covers firms from manufacturing and service sectors and is conducted as a mail survey with the option to respond online.

In the MIP 2019, firms were asked whether they introduced a product or process innovation within the last three years (between 2016 and 2018) and for the total amount spent on innovation activities in the last year (2018). We consider a firm that stated, it introduced a product innovation within the considered time frame as a product innovator and a firm that stated that it introduced a process innovation within the considered time frame as a process innovator. A firm is an innovator if it introduced at least one of both. Every firm that spent financial resources on innovation—independent of the magnitude—is regarded as a firm with innovation expenditures. Our initial sample consists of 13,747 firms from the MIP 2019. We

merge these firms with the Mannheim Enterprise Panel (MUP, see [33]), which consists of more than 3.2 million economically active firms, to receive information about the firms' website addresses. The MUP serves as a sampling frame for surveys like the MIP and, e.g., contains firm-level information on turnover, number of employees and sector affiliation. Only 54 percent of firms in our sample can be assigned to website addresses, as we limit ourselves to quality-assured observations. In total, we have 6,368 firms with information on the website address and at least one innovation indicator. We extract website content by applying the ARGUS web-scraper, which allows us to collect texts as well as hyperlinks to other websites (for a detailed description of the ARGUS web-scraper see [13, 34]). Firm websites were first scraped in September 2018 to collect texts, then again in January 2019 for adding hyperlinks. We scraped a third time in October 2019 to add information about technical features, e.g., capturing the existence of firm websites for mobile end user devices. The maximum limit of scraped subpages per website is set to 50, otherwise the amount of data would become too large. We consider this to be a sufficient number, as the median number of subpages in the MUP is 15 (see [13]) and only 1.5 percent of all firms in our subsample have 50 or more subpages. Moreover, the scraping program is set to prefer subpages with shorter website addresses because we assume these subpages include more important information about the firm. Also, ARGUS is set to prefer websites in German language. Hence, when we calculate the share of different languages on a website we expect a small bias. However, since only a few firms exceed the subpage limit, we assume this bias to be negligible. While scraping the data, especially while collecting meta information features, we received several error messages. Furthermore, we only use observations for which all features are non-missing. If, for example, a meta information feature is not available the observation will not be used for training or testing with other feature sets. Therefore, after the entire data collection process, we end up with 4,487 firms in our sample when predicting product innovators and innovators, 4,484 firms when predicting process innovators and 1,893 when predicting whether a firm has innovation expenditures (Table 1). There are three observations more for product innovators than for process innovators. Since these three observations are all product innovators, they are also in the innovator sample.

Additionally, a random sample of 32,276 website addresses of firms not included in the MIP is drawn from the MUP and scraped with the ARGUS web-scraper using the same settings as for the MIP sample. The sample is used for topic modelling. We train a topic model on a separate sample for two reasons. First, it allows to include more data points. Second, it ensures that no observation used for calibrating topics is considered for evaluating Random Forest models. Hence, it prevents data leakage. The sample is hereinafter referred to as the LDA sample.

As we need to exclude a large share of observations due to missing values in our MIP sample, we cannot rule out a selection bias. Also, firms from certain industries and smaller firms are less likely to have a website and may therefore be underrepresented. In machine learning,

**Table 1. Summary statistics for product innovators, process innovators, innovators as well as firms with innovation expenditures.**

| Variable | Definition | N | Mean | SD | Min | Max |
|---|---|---|---|---|---|---|
| Product innovators | 1: If firm is a product innovator<br>0: Otherwise | 4,487 | 0.39 | 0.49 | 0 | 1 |
| Process innovators | 1: If firm is a process innovator<br>0: Otherwise | 4,484 | 0.52 | 0.50 | 0 | 1 |
| Innovators | 1: If firm is a product or / and process innovator<br>0: Otherwise | 4,487 | 0.61 | 0.49 | 0 | 1 |
| Innovation expenditures | 1: If firm innovation expenditures were reported<br>0: Otherwise | 1,893 | 0.39 | 0.49 | 0 | 1 |

adverse selection might lead to two issues: It could cause that our model is better fitted for groups that are overrepresented in our sample and it could induce that the class correlated with the overrepresented group is predicted more often. To identify whether a potential selection bias exists, we analyze how the sample distribution changes with respect to the number of employees and industry sectors, when excluding observations with missing information (see S1 and S2 Figs).

Except for "transportation and post" (sector 15), we do not see a notable change in the distribution of firms that could be linked to a severe selection bias.

To capture website characteristics, we apply several methods to generate features like a keyword search and natural language processing as well as an analysis of hyperlinks (network analysis methods). We use Python as programming language for calculating our features and for training our Random Forest models. For an overview of feature sets see Table 2.

## 3.1 Text-based features

Information from website texts is analyzed, as it might be related to a firm's innovation status for the following reasons: Presumably, most firms are using their websites to inform customers about new products or services and might mention whether their product is new or innovative, i.e., it is likely that innovative firms use particular innovation-related words. Information about process innovations can also be detected and used if reported on the website. Moreover, a firm might report that it uses a recently emerging technology like blockchain, 3D printing or augmented reality (for an overview of recently emerging technologies, see S1 Text).

**Table 2. Features related to text, meta information and network measures.**

| | |
|---|---|
| *Text-based features* | |
| 1) Textual content | Term-document matrix with the 5,000 most frequent words (TF-IDF applied). |
| 2) Emerging technologies | Dummy variable that measures whether a technology of Wikipedia's list of emerging technologies appears on a firm's website. |
| 3) Latent patterns | Topic-document probabilities of 150 topics generated by the LDA approach. |
| 4) Topic popularity index | The sum of LDA topic probabilities per document. Each probability is weighted with the relative frequency of its appearance in the entire LDA sample. |
| 5) International orientation | Share of subpages in English language and the share of all other non-German subpages in all subpages. |
| 6) Share of numbers | The share of numbers in website text (characters). |
| 7) Flesch-reading-ease score | Numerical metric assessing readability of texts. |
| *Meta information features* | |
| 8) Website size | Number of subpages on a website, total amount of characters on a website. |
| 9) Loading time | The time from sending a request (http/https) to a webserver (to get the start page of a website) until the arrival of the response (in ms). |
| 10) Mobile version | Dummy variable that is one if a version for mobile end user devices exists and zero otherwise. |
| 11) Domain purchase year | The year of the first entry at web.archive.org. |
| *Network features* | |
| 12) Centrality | The total number of incoming, the total number of outgoing hyperlinks as well as the PageRank centrality. |
| 13) Social media | Number of hyperlinks to Facebook, Instagram, Twitter, YouTube, Kununu, LinkedIn, XING, GitHub, Flickr, and Vimeo. |
| 14) Bridges | Number of bridges a firm is part of in the hyperlink network. |

Hence, an emerging technology term might appear on a firm's website and if so it is likely that the firm can be considered as innovative, at least on an incremental level, as it makes use of technologies that are fairly new. Additionally, there might be latent patterns on a website that reveal a firm's innovation status, these latent patterns can be captured by the LDA topic modelling approach as successfully shown in [21]. Furthermore, innovative firms might follow some general technological trends like the digital transformation. As these technological trends are quite general, LDA topics related to these trends might appear quite often on firm websites. To capture this, we construct a topic popularity index that indicates the distribution of popular and less popular topics on a website.

We additionally analyze the following text-based metrics: Languages that appear on a website might relate to the export status of a firm and this could provide information about a firm's innovation status because the export status is linked to firm-level innovation (e.g., [35–37]). Also, we test whether the share of numbers in all string characters (text) as well as the text complexity measured by the Flesch-reading-ease score [38] differ between innovative and non-innovative firms.

## 3.2 Meta information features

Second, meta information of firm websites (see Table 2) might allow to distinguish innovative from non-innovative firms. For example, the website size might help to predict a firm's innovation status. Large firms are more likely to be innovative [10]. As the number of subpages of a website correlates with the number of employees of a firm [13], the size of a website might provide information about whether a firm introduced an innovation. Also, the technological properties of a website could be relevant. Innovative firms might have a better technical knowledge and are able to apply more technologically advanced features on their websites. For example, the loading time of a website could be faster and a mobile version might be more often available when firms are more technologically advanced. However, there might be some noise because the loading time may also be short if the website is relatively simple.

Another potentially relevant feature is the age of a website, i.e., the domain purchase year, as it might relate to the actual firm age. One has to consider, however, that this relationship is unlikely to be linear. On the one hand, a website that is fairly new might indicate a start-up with an innovative idea. On the other hand, having a very old website means the firm has adopted this new technology very early. This could also relate to a more technologically advanced, hence, innovative firm.

## 3.3 Network features

Third, hyperlinks between websites (see Table 2) might also help to identify the firm-level innovation status. Firms that have more business relationships with other firms or are more relevant according to centrality measures might be better informed and know earlier about new profitable applications. Hence, firms with more relationships to other firms could be more likely to be innovative. Moreover, innovation projects are often realized in cooperation with other firms (e.g., [39]). Thus, patterns in firm-level cooperation are expected to be of interest. A firm that connects (or bridges) different network parts is usually relevant and its removal will decompose the network. Lastly, [40] show that a firm's use of the social networking site Facebook is linked to product innovations. Hence, the use of social media might reveal information about a firm's innovation status, as well.

Our study analyzes whether the three groups of features differ in their performance when predicting a firm's innovation status. A more detailed description of the feature generation can be found in S2 Text.

**Table 3. Descriptive statistics for selected variables.**

| | Group-specific means | | | | | | | | | | | |
|---|---|---|---|---|---|---|---|---|---|---|---|---|
| | Product innovator | | | Process innovator | | | Innovator | | | Innovation expend. | | |
| Feature (Variable name) | Yes | No | P-val. | Yes | No | P-val. | Yes | No | P-val. | Yes | No | P-val |
| Text-based features | | | | | | | | | | | | |
| Emerging technology term (*emerging_tech*) | 0.18 | 0.07 | 0.00 | 0.15 | 0.07 | 0.00 | 0.15 | 0.05 | 0.00 | 0.19 | 0.06 | 0.00 |
| Percentage of English language (*english_language*) | 0.16 | 0.10 | 0.00 | 0.14 | 0.10 | 0.00 | 0.14 | 0.09 | 0.00 | 0.17 | 0.08 | 0.00 |
| Percentage of other language (*other_lang*) | 0.02 | 0.02 | 0.45 | 0.02 | 0.02 | 0.25 | 0.02 | 0.02 | 0.74 | 0.02 | 0.02 | 0.30 |
| Topic popularity index (*pop_score*) | 34.64 | 34.35 | 0.36 | 34.78 | 34.11 | 0.03 | 34.68 | 34.13 | 0.08 | 35.07 | 33.82 | 0.01 |
| Share of numbers (*share_numbers*) | 0.025 | 0.028 | 0.00 | 0.025 | 0.028 | 0.00 | 0.026 | 0.028 | 0.00 | 0.027 | 0.027 | 0.97 |
| Flesch-reading-ease score (*flesch_score*) | 40.09 | 41.22 | 0.01 | 40.54 | 41.03 | 0.26 | 40.47 | 41.26 | 0.09 | 39.28 | 41.28 | 0.01 |
| Meta information features | | | | | | | | | | | | |
| Website size: Length (*text_length*) | 75269.35 | 56746.84 | 0.00 | 71629.95 | 55685.73 | 0.00 | 71193.63 | 52859.37 | 0.00 | 75334.75 | 52462.63 | 0.00 |
| Website size: Nr. of pages (*nr_subpages*) | 30.37 | 24.65 | 0.00 | 28.75 | 24.87 | 0.00 | 28.92 | 23.75 | 0.00 | 31.23 | 23.58 | 0.00 |
| Loading time (*load_time*) | 0.57 | 0.55 | 0.69 | 0.51 | 0.60 | 0.25 | 0.55 | 0.57 | 0.76 | 0.51 | 0.49 | 0.57 |
| Mobile version (*mobile_version*) | 0.76 | 0.70 | 0.00 | 0.76 | 0.68 | 0.00 | 0.75 | 0.67 | 0.00 | 0.73 | 0.69 | 0.06 |
| Domain purchase year (*domain_purchase_year_proxy*) | 2004.22 | 2004.98 | 0.00 | 2004.42 | 2004.96 | 0.00 | 2004.37 | 2005.17 | 0.00 | 2004.38 | 2005.01 | 0.01 |
| Network features | | | | | | | | | | | | |
| Outgoing hyperlinks (*outgoing_links*) | 15.93 | 12.95 | 0.00 | 15.18 | 12.97 | 0.00 | 15.19 | 12.46 | 0.00 | 16.23 | 12.38 | 0.00 |
| Incoming hyperlinks (*incoming_links*) | 14.78 | 5.22 | 0.00 | 13.24 | 4.30 | 0.00 | 12.11 | 4.09 | 0.00 | 12.09 | 3.70 | 0.00 |
| Use of social media (*social_media*) | 1.62 | 1.02 | 0.00 | 1.51 | 0.98 | 0.00 | 1.47 | 0.92 | 0.00 | 1.62 | 0.91 | 0.00 |
| PageRank centrality (*pagerank_index*) | $2*10^{-6}$ | $1*10^{-6}$ | 0.00 | $2*10^{-6}$ | $1*10^{-6}$ | 0.00 | $1*10^{-6}$ | $1*10^{-6}$ | 0.00 | $1*10^{-6}$ | $1*10^{-6}$ | 0.01 |
| Bridges (*bridge_index*) | 0.43 | 0.26 | 0.01 | 0.38 | 0.28 | 0.05 | 0.37 | 0.27 | 0.04 | 0.31 | 0.27 | 0.35 |
| Number of observations | 4,487 | | | 4,484 | | | 4,487 | | | 1,893 | | |

Source: MIP 2019 and web-scraped data; Own calculations. All variables were rounded to the second decimal place except PageRank centrality, which was rounded to the sixth decimal place and share of numbers which was rounded to the third decimal place.

## 4 Descriptive analysis

The descriptive statistics for our predictor variables are presented in this section. Table 3 shows mean values for innovative and non-innovative firms as well as p-values regarding the difference of both means for selected features.

Differences exist for most variables. Looking at 'text' features, innovative firms are more likely to mention an emerging technology term and have more subpages in English language. The share of subpages in other languages, however, does not show any significant difference between both groups. Differences are also small for the share of numbers, our topic popularity index and for the Flesch-reading-ease score, but the deviation is statistically significant for some forms of innovation activity.

The descriptive statistics for 'meta' features show that innovative firms have larger websites with respect to the number of subpages as well as with respect to the number of characters. The loading time is slightly faster for process innovators and innovators, but not for product innovators and firms with innovation expenditures. However, differences are not statistically significant. The first occurrence on web.archive.org is significantly later for non-innovative firms indicating their domain purchase year, i.e., website age is slightly lower. Additionally, non-innovative firms have less often a version of their website for mobile end user devices. Looking at 'network' features, significant differences also exist for outgoing and incoming hyperlinks as well as for hyperlinks to social media websites. Innovative firms have on average

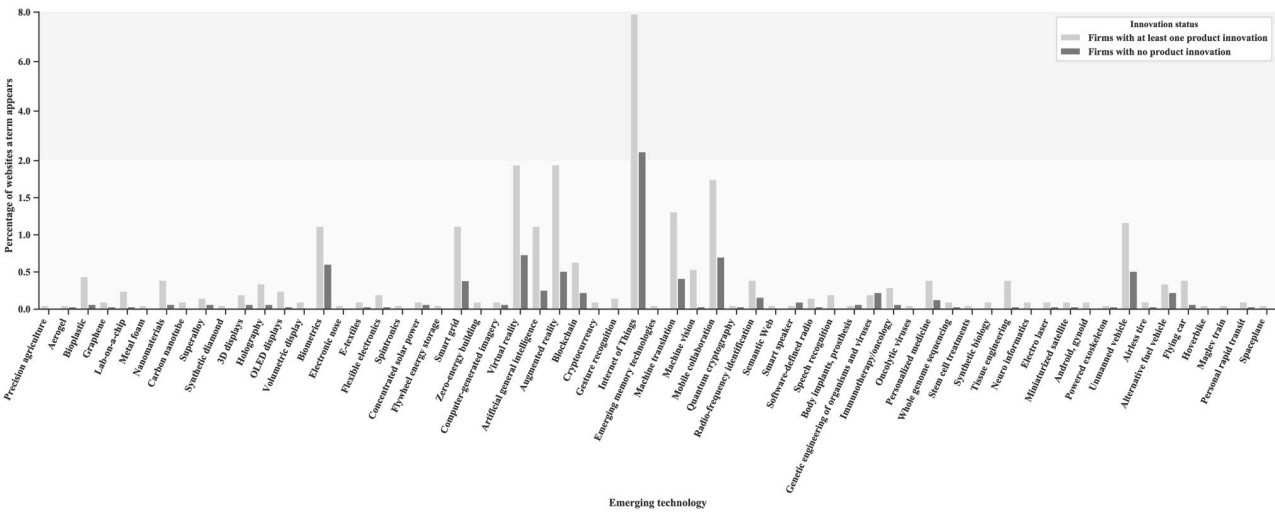

**Fig 1. Average occurrence of different emerging technology terms on firm websites with and without product innovations.** For instance, the emerging technology term *virtual reality* appears on nearly 2 percent of all product innovator websites, but only on approximately 0.75 percent of all non-product innovator websites. Emerging technology terms not appearing on firm websites are not illustrated. The y-axis has a scale break at 2 percent.

more hyperlinks. Moreover, the difference is larger for incoming than for outgoing or social media hyperlinks. Additionally, innovative firms also are significantly more important in firm networks looking at the PageRank centrality. The statistical significance of differences regarding the bridge index is, however, limited to the form of innovation activity. In summary, Table 3 confirms previous assumptions. Innovative firms seem more likely to apply emerging technologies, to have more technically advanced websites and to be better connected with each other according to most network indicators.

Fig 1 shows the average occurrence of different emerging technology terms on a firm website with respect to product innovation. The emerging technology terms differ strongly in their likelihood of occurrence. The emerging technology term *Internet of Things* is the most likely to occur. It appears on more than 8 percent of all product innovator websites and only on less than 2 percent of all non-product innovator websites. Also, terms relating to different machine learning applications, *biometrics*, *blockchain* technology and *mobile collaboration* appear relatively often. Moreover, for nearly every emerging technology term it is more likely to appear on a product innovator website than on a non-product innovator website. This result is the same for all innovation indicators.

Table 4 shows the ten most innovation-relevant LDA topics. The highest average value of Pearson correlation coefficients for all four innovation indicators and the document-topic probabilities is used to identify the most relevant LDA topics. The topics are sorted in descending order. LDA topic 98, which relates according to its keywords to research & development, has a positive and by far the strongest relationship to innovation. Also, LDA topic 35, which relates to ICT infrastructure, has a comparatively strong positive correlation with our innovation indicators. Among the top 10, the LDA topics 20 (tourism), 120 (consulting & customer support) and 23 (family business & craftsmanship) have the weakest correlation. Moreover, the correlation is negative.

Fig 2 also relates to the ten most innovation-relevant LDA topics. It shows for every topic the average share in a document for innovative and non-innovative firms. The figure reflects the results presented in Table 4. The selected topics considerably differ between innovative

**Table 4. Content of the LDA topics with the strongest relationship to MIP-based innovation indicators.**

| Topic number | Content | Translated | Top words | Correlation* |
|---|---|---|---|---|
| LDA topic 98 | Research & development | yes | 'company' 'customer' 'development' 'to develop' 'department' 'employee' 'partner' 'project' 'successful' | positive (0.15) |
| LDA topic 35 | ICT infrastructure | yes | 'system' 'software' 'data centers' 'server' 'version' 'support' 'date' 'windows' 'automatic' 'document' | positive (0.10) |
| LDA topic 65 | Construction | yes | 'to build' 'project' 'new building' 'architect' 'planning' 'renovation' 'reconstruction' 'construction' 'to plan' 'architecture' | negative (-0.09) |
| LDA topic 134 | Business software | no | 'array' 'value' 'news' 'office' 'paket' 'error' 'data' 'page' 'SAP' 'search' | positive (0.08) |
| LDA topic 7 | Product experience | no | 'centro' 'company' 'best' 'use' 'experience' 'world' 'please' 'product' 'may' 'find' | positive (0.08) |
| LDA topic 41 | Common terms | yes | 'and' 'far' 'to take place' 'to put' 'frame' 'that' 'information' 'total' 'receive' 'department | negative (-0.07) |
| LDA topic 5 | Carpentry | yes | 'to tile' 'woods' 'to lay' 'laminate' 'tile' 'to put' 'material' 'stairs' 'floor' 'to glaze' | negative (-0.07) |
| LDA topic 20 | Tourism | yes | 'region' 'city' 'to be located' 'to offer' 'museum' 'old' 'historical' 'nature' 'tour' 'landscape' | negative (-0.06) |
| LDA topic 120 | Consulting & customer support | yes | 'pleased' 'to offer' 'customer' 'to advise' 'individual' 'consulting' 'available' 'question' 'competent' 'to find' | negative (-0.06) |
| LDA topic 23 | Family business & craftsmanship | yes | 'company' 'to operate' 'visit' 'to stand' 'roofing' 'Michael' 'son' 'specialize' 'work' | negative (-0.06) |

* Measured by the average of all Pearson correlation coefficients between the average topic share per document and each innovation indicator.

and non-innovative firms. Also, relationships are constant, e.g., if a topic has a larger share on product innovator than on non-product innovator websites, it will also be relatively stronger represented on process innovator websites. Nonetheless, differences between innovation indicators exist. Average topic share differences diverge between indicators and are larger when

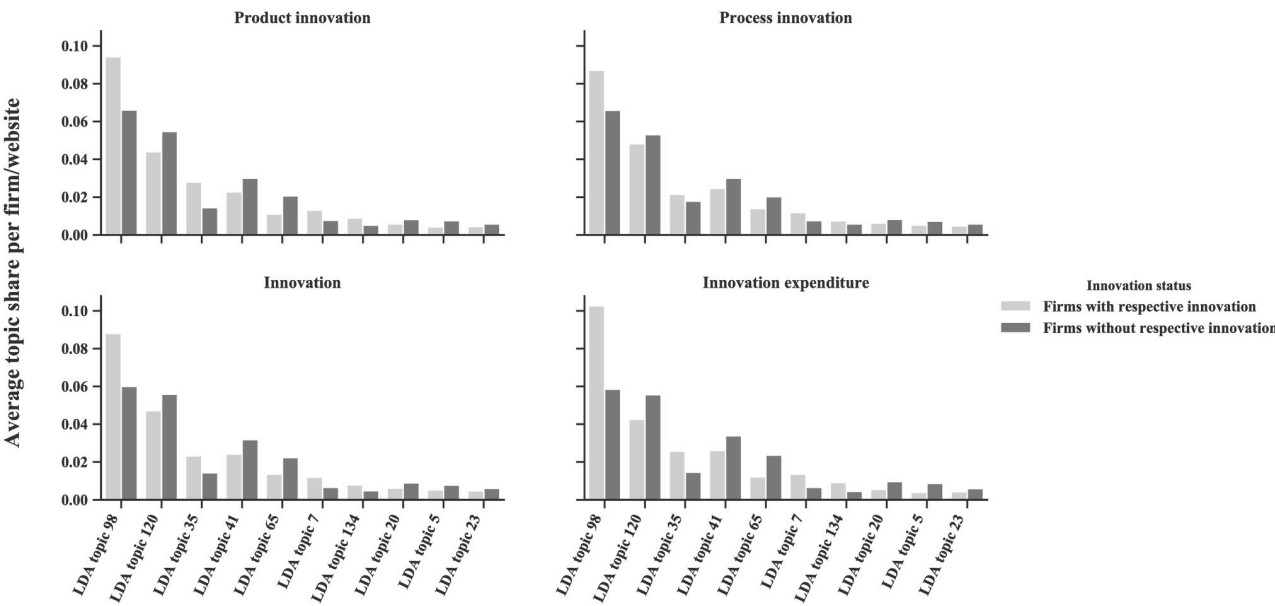

**Fig 2. Differences in the topic share of the top 10 topics with the strongest correlation with MIP-based innovation indicators on average.** For instance, the LDA topic 98 has an average share of 10 percent in a document if a firm has innovation expenditure, compared to merely 6 percent if a firm does not have innovation expenditure.

considering firms' innovation expenditures than when taking product or process innovators into account.

## 5 Methodology

The objective of our work is the identification of website characteristics that allow predicting firm-level innovation activities. For this purpose, we integrate the described features as predictor variables in Random Forest classification models [41, 42]. For each of our feature sets ('text', 'meta' and 'network' features) as well as for all features jointly a separate Random Forest model is fitted. We use the Python package *scikit-learn* for the exercise. The Random Forest algorithm is an ensemble method used for classification or regression tasks. Like any other machine learning algorithm as defined in [43], it uses past experience (in our case survey data) to learn how to perform predictions. The Random Forest algorithm makes its decision based on the modus or mean of a multitude of decorrelated decision trees. Each tree is built based on bootstrapped samples of training data. By splitting the data at nodes into branches that are more "pure" with respect to the target variable, the algorithm learns to improve. We chose the Random Forests algorithm because it has the advantage that it allows for the calculation of feature importances, while providing high predictive power and enabling the consideration of complex interactions.

For instance, feature importances can be derived by means of the MDI [44], which is a measure based on a split criterion that is used to build single decision trees (for an overview of different split criterion measures see [45]). In our study, we use the "decrease in impurity" as a split criterion. We calculate at each node to what extent a particular split will decrease the impurity of respective child nodes. The combination of a variable and splitting value that leads to the best split, weighted by the number of observations within each child node, will be selected. A formal description of the "decrease in impurity" is given by Eq (1). $i(t)$ measures impurity at the node level, which is in our case indicated by the Gini impurity index. $t$ is a node within one tree and $s$ is a split at a certain value of a variable. $N_x$ is the number of samples reaching node $x \in \{t, t_L, t_R\}$. Lastly, if $t$ is the parent node, $t_L$ is the left child node and $t_R$ is the right child node for the split $s$ at node $t$. The split $s$ for node $t$ that maximizes $\Delta i(s, t)$ is iteratively chosen.

$$\Delta i(s, t) = i(t) - N_{t_R}/N_t * i(t_R) - N_{t_L}/N_t * i(t_L) \qquad (1)$$

Feature importance is then derived by the sum of "decreases in impurity" of a single variable divided by the sum of "decreases in impurity" of all features used to build the tree. The value is additionally averaged over all trees in the forest and again normalized so that all values sum up to one. If multiple variables will lead to similar impurity decreases at one node, only one variable is selected for splitting. Hence, (multi-)collinearity of features can bias feature importance. This issue can be illustrated by the following. In this example, the same variable is included twice in a model. When choosing a variable for splitting, the model can randomly choose between the two and the feature relevance is thus divided between both variables.

To evaluate the performance of collected website characteristics, we use a baseline model. A random coin toss model based on the sample distribution is chosen. A baseline model works as a benchmark to assess the performance of more complex solutions, i.e., it helps to analyze whether a trained model performs better than a random prediction. To estimate whether we achieve considerable improvements in comparison to baseline predictions, we perform a McNemar test [46]. Assuming a chi-squared frequency distribution, the McNemar test measures if predictions from two machine learning models significantly disagree with each other as illustrated in Eq (2). *RF* captures the number of observations misclassified by a fitted

Random Forest model, but not by the baseline model. *BL* captures the number of observations misclassified by the baseline model, but not by a fitted Random Forest model.

$$\chi^2 = \frac{(RF - BL)^2}{(RF + BL)} \tag{2}$$

If a model including a distinct feature set significantly disagrees with baseline predictions according to the McNemar test and evaluation metrics show superior values, we consider this feature set to be relevant for the prediction of firm-level innovation activity.

To further evaluate and compare models, we use the metrics "area under the curve" (AUC), accuracy, improvement of accuracy in comparison to the baseline model, precision, recall, and the F1-score for positive as well as negative observations [47].

$$\text{false positive rate} = \frac{FP}{(FP + TN)} \tag{3}$$

$$\text{true positive rate (recall for the positive class)} = \frac{TP}{(TP + FN)} \tag{4}$$

The AUC can be explained as follows. The formulas listed in Eqs (3) and (4) are based on the number of false positive predictions (FP), capturing non-innovative firms wrongly predicted as innovative; true positive predictions (TP), capturing innovative firms correctly predicted as innovative; false negative predictions (FN), capturing innovative firms wrongly predicted as non-innovative; and true negative predictions (TN), capturing non-innovative firms correctly predicted as non-innovative. The receiver operating characteristic curve (ROC) is a graphical illustration of a binary classifier performance. For different classification thresholds, the "false positive rate" is plotted against the "true positive rate" and the AUC value is an approximation of the area below the ROC. Accordingly, the AUC value is the probability that a randomly chosen innovative firm is assigned a higher probability of being innovative than a randomly chosen non-innovative firm. Usually, AUC values above 0.7 are considered as acceptable [48].

For the other metrics a classification threshold has to be set. The classification threshold is also called cut-off value and refers to the transformation of the regression output to a binary classification. Different cut-off values can be chosen if for example "false negatives" are considered more costly than "false positives" or if certain metrics need to be optimized. We select 0.5 as a cut-off value for all fitted models, because this value is most commonly used and we do not prefer one metric or class over the other.

$$\text{precision for positive class} = \frac{TP}{(TP + FP)} \tag{5}$$

$$\text{precision for negative class} = \frac{TN}{(TN + FN)} \tag{6}$$

$$\text{true negative rate (recall for the negative class)} = \frac{TN}{(FP + TN)} \tag{7}$$

Formal definitions of precision for innovative and for non-innovative firms are illustrated in Eqs (5) and (6). Recall for innovative and for non-innovative firms is measured by the "true positive rate" or "true negative rate" as illustrated in Eqs (4) and (7). Precision measures, for instance, the share of correctly classified innovative firms in all firms classified as innovative,

while recall measures the fraction of innovative firms that have been correctly identified as innovative.

$$accuracy = \frac{(TP + TN)}{(TP + TN + FP + FN)} \tag{8}$$

$$F1 - score_{P,N} = 2 * \left( \frac{(Precision_{P,N} * Recall_{P,N})}{(Precision_{P,N} + Recall_{P,N})} \right) \tag{9}$$

Accuracy and F1-score are presented in Eqs (8) and (9). Accuracy measures the share of correct predictions in all predictions. The F1-score captures the harmonic mean between precision and recall for positive (P) and negative (N) observations, respectively. Respective baseline outcomes of accuracy, F1-scores as well as precision and recall for our different innovation activity indicators are presented in Table 5 in Section 6. The random coin toss model assumes a fixed chance of being innovative (based on the sample mean). Hence, results do not change when the threshold is varied and therefore the AUC value is not displayed for baseline outcomes.

**Table 5. Results for Random Forest classification models using different feature sets and target variables.** Evaluation metrics are presented for the test sample.

| Feature sets | | | | | Accuracy | | F1-Score | | Precision | | Recall | | McNemar | |
|---|---|---|---|---|---|---|---|---|---|---|---|---|---|---|
| Baseline | Text | Meta | Network | AUC | Value | Δ | Positive | Negative | Positive | Negative | Positive | Negative | P-values | Support |
| Product innovators | | | | | | | | | | | | | | |
| x | | | | - | 0.53 | - | 0.39 | 0.61 | 0.39 | 0.61 | 0.39 | 0.61 | - | 1,122 |
| | x | | | 0.72 | 0.69 | 0.16 | 0.47 | 0.78 | 0.69 | 0.69 | 0.35 | 0.90 | 0.00 | 1,122 |
| | | x | | 0.66 | 0.64 | 0.11 | 0.37 | 0.75 | 0.59 | 0.66 | 0.27 | 0.88 | 0.00 | 1,122 |
| | | | x | 0.65 | 0.66 | 0.13 | 0.30 | 0.77 | 0.72 | 0.65 | 0.19 | 0.95 | 0.00 | 1,122 |
| | x | x | x | 0.73 | 0.70 | 0.17 | 0.49 | 0.79 | 0.71 | 0.70 | 0.37 | 0.90 | 0.00 | 1,122 |
| Process innovators | | | | | | | | | | | | | | |
| x | | | | - | 0.50 | - | 0.52 | 0.48 | 0.52 | 0.48 | 0.52 | 0.48 | - | 1,121 |
| | x | | | 0.62 | 0.59 | 0.09 | 0.63 | 0.54 | 0.59 | 0.59 | 0.67 | 0.50 | 0.00 | 1,121 |
| | | x | | 0.60 | 0.57 | 0.07 | 0.64 | 0.46 | 0.57 | 0.58 | 0.74 | 0.39 | 0.01 | 1,121 |
| | | | x | 0.59 | 0.57 | 0.07 | 0.62 | 0.52 | 0.58 | 0.56 | 0.66 | 0.48 | 0.01 | 1,121 |
| | x | x | x | 0.63 | 0.60 | 0.10 | 0.64 | 0.55 | 0.60 | 0.60 | 0.68 | 0.52 | 0.00 | 1,121 |
| Innovators | | | | | | | | | | | | | | |
| x | | | | - | 0.52 | - | 0.60 | 0.40 | 0.60 | 0.40 | 0.60 | 0.40 | - | 1,122 |
| | x | | | 0.67 | 0.63 | 0.11 | 0.75 | 0.30 | 0.63 | 0.59 | 0.91 | 0.20 | 0.00 | 1,122 |
| | | x | | 0.64 | 0.62 | 0.10 | 0.74 | 0.33 | 0.64 | 0.56 | 0.88 | 0.23 | 0.00 | 1,122 |
| | | | x | 0.62 | 0.60 | 0.08 | 0.75 | 0.00 | 0.60 | 0.00 | 1.00 | 0.00 | 0.00 | 1,122 |
| | x | x | x | 0.68 | 0.63 | 0.11 | 0.75 | 0.31 | 0.64 | 0.59 | 0.91 | 0.21 | 0.00 | 1,122 |
| Innovation expenditures | | | | | | | | | | | | | | |
| x | | | | - | 0.54 | - | 0.36 | 0.64 | 0.36 | 0.64 | 0.36 | 0.64 | - | 474 |
| | x | | | 0.74 | 0.73 | 0.19 | 0.55 | 0.80 | 0.68 | 0.74 | 0.47 | 0.88 | 0.00 | 474 |
| | | x | | 0.67 | 0.65 | 0.11 | 0.33 | 0.76 | 0.53 | 0.67 | 0.24 | 0.87 | 0.00 | 474 |
| | | | x | 0.65 | 0.67 | 0.13 | 0.25 | 0.79 | 0.68 | 0.67 | 0.16 | 0.96 | 0.00 | 474 |
| | x | x | x | 0.75 | 0.72 | 0.18 | 0.55 | 0.80 | 0.67 | 0.74 | 0.47 | 0.87 | 0.00 | 474 |

Source: MIP 2019 and web-scraped data; Own calculations. Numerical values are rounded. The baseline values are calculated assuming perfect knowledge about the test sample distribution, which means that the test sample mean is used for predictions. P-values relate to the significance level at which a model disagrees with its baseline model according to the McNemar test for 10.000 baseline prediction rounds. The significance levels are based on mean values.

To control for overfitting, we analyze the performance by using out-of-sample predictions. Accordingly, we do not evaluate the models' performance with the observations that are already used for learning. The data is split into a training sample (for fitting models) and a test sample (for evaluating models). To be more precise, the test sample is a "hold-out" sample and therefore never used for model training.

The training sample consists of 75 percent and the test sample consists of 25 percent of our observations. In the supervised learning context, this is a common partitioning method. It constitutes a trade-off between the generalization of the model and the validity of the evaluation. We also apply a grid-search to tune the hyperparameters of all our models [42] on our training sample. We explore the hyperparameter space for the 'number of trees' (100, 500, 1,000, and 1,500), 'maximum tree depth' (50, 100, 150, and 200), and 'minimum impurity decrease' (0.01, 0.001). For all other hyperparameters we use default values provided by *scikit-learn*.

This leads to 32 different hyperparameter combinations for every model. For each hyperparameter combination in our grid-search a 5-fold cross-validation is performed. The k-fold cross-validation belongs to the non-exhaustive cross-validation methods. It is a technique to assess generalizability of machine learning models to new data, detect overfitting and potential sample biases. The data is split into k subsets so that $100 - (100/k)$ percent of the data is used for training the model and $100/k$ percent for validation. In each of the k iterations a different training and validation data set is used.

Considering all models fitted in the cross-validated grid-search, we choose the model with the highest AUC value. The selected model is then evaluated on the test sample.

To ensure the reproducibility of our study, we fixate the random seed when necessary. The random seed influences the model performance to some extent, e.g., observations are assigned to the train or test sample based on the random seed.

## 6 Results

In this section, the predictions of MIP-based innovation indicators using a Random Forest classification approach are described. Table 5 shows evaluation metrics for all baseline as well as fitted models. We analyze four different innovation indicators (four target variables), which we predict based on three different subsets of features as well as their union (four different groups of features). Accordingly, we train 16 Random Forest models.

Looking at product innovators, the highest AUC score (0.73) is realized with 'all' features. The baseline accuracy is 0.53. The largest increase can be observed for the 'all' feature model (17 percentage points). Text-based features alone, however, lead to an increase of 16 percentage points. Moreover, 'network' and 'meta' features have a relatively weak impact. They just lead to improvements of 13 and 11 percentage points, respectively. This indicates that a large share of predictive power results from website text. The baseline F1-score for product innovators is 0.39 and for non-product innovators it is 0.61. Hence, the sample is slightly imbalanced towards non-product innovators and chances of randomly predicting this class correctly are higher. Furthermore, the F1-scores show a similar result to other metrics. Only the 'text' and the 'all' feature model improve F1-scores notably. When solely applying 'meta' or 'network' features, F1-scores for innovative firms are even worse than the baseline performance. Precision values do not considerably differ between innovative and non-innovative firms and are always higher than the baseline prediction. Moreover, there is a comparatively large increase in precision for innovative firms.

In contrast, there is a great difference between both classes with respect to recall values. For innovative firms, recall values of fitted models are always worse than those of the baseline prediction. For non-innovative firms, the recall fluctuates between 88 and 95 percent.

Our evaluation metrics for models predicting process innovators have predominantly lower values than those predicting the product innovator status. Nonetheless, fitted models show for nearly all evaluation metrics better results than the process innovator baseline model and the McNemar test also confirms a significant difference. Hence, website characteristics still improve predictions. The best performance, in terms of accuracy, is reached by our 'all' feature model and leads to a performance increase of 10 percentage points. Moreover, 'meta' and 'network' features perform slightly worse than 'text' features.

The performance for innovators is slightly better than for process innovators in terms of AUC and accuracy. As the sample is slightly imbalanced towards innovators, this performance difference, however, is also partly related to different baseline values. Furthermore, similar to product innovators, we see remarkably higher AUC values of models including 'text' features. However, considering all other evaluation metrics 'meta' features perform very similar to 'text' features. Looking at F1-scores, predictions for the negative class always perform worse than the baseline model. In particular, the prediction solely based on 'network' features leads to zero F1-scores. This means the model predicts for every firm a likelihood that a firm is innovative larger than 0.5, which implies that the model always predicts the majority class. This is known as zero rule prediction. For applying this rule, the information included in our baseline model is sufficient. In this regard, 'network' features do not provide information gains for innovators. Looking at precision and recall (and not considering the 'network' feature model), we find general improvements for innovative firms in comparison to the baseline model. For non-innovative firms, we only find improvements in precision. Recall values, however, are very low and worse than in the baseline model.

Even though the number of observations is the smallest, the predictive performance as well as the performance increase for firms with innovation expenditures is the highest in terms of AUC and accuracy. Looking at the 'all' feature model, firms with innovation expenditures can be predicted with an AUC value of 75 percent and an accuracy of 0.72 percent, which corresponds to an accuracy increase of 18 percentage points. The model solely based on 'text' features performs even slightly better than the 'all' feature model considering accuracy. Besides, values of all other evaluation metrics are always better than random for the 'text' and 'all' feature model. Both models only using 'network' or 'meta' features show also strict improvements in accuracy and precision, but F1-scores and recall are partly worse than the baseline model.

Furthermore, the McNemar test confirms that all fitted models significantly disagree with baseline predictions. The divergence is always highly significant (p-values are below 0.001), except for models that predict process innovators with either 'meta' or 'network' features, which are significant at the 0.01-level. This may be due to the fact that both feature sets as well as models predicting process innovators perform relatively worse. Hence, the difference to baseline predictions is especially low when combining both. It is also noteworthy that even though the McNemar test is significant, it does not necessarily mean that the model is strictly better than the baseline model. Key evaluation metrics also have to show predominately superior values. We want to highlight one example here. The Random Forest model that predicts innovators using 'network' features has a large share of inferior values in comparison to baseline predictions. It uses the zero rule for its prediction. Accordingly, it significantly disagrees with the baseline model as it uses another decision rule. However, the fitted model is not strictly better, because its decision is also solely based on the sample mean and the fitted model is not learning sufficiently from the provided features as the evaluation metrics show.

Lastly, we want to note that we do not find a particular combination of hyperparameters across innovation indicators and feature sets that is always selected by the grid-search algorithm. However, preferred 'number of trees', 'maximum tree depth', and 'minimum impurity decrease' do exist across feature sets and target variables. For the 'number of trees' 1000 and

1500 are mostly chosen. The most dominant 'maximum tree depth' values are 50 and 100. Moreover, a 'minimum impurity decrease' of 0.001 is more popular than 0.01. For more details see S3 Table.

To analyze the robustness of presented results, we re-estimate the 'all' feature model for each indicator using all possible combinations of splits between the training and test sample from 0.1/0.9 to 0.9/0.1 (in steps of 0.01). The corresponding change of respective AUC values with respect to an increasing training sample is displayed in S3 Fig. We find that AUC values for product innovators, process innovators and innovators increase until a training sample size of 0.6 and then stay roughly constant at levels pointed out in Table 5. Hence, AUC values seem robust with respect to the sample split if a sufficiently large training sample size is reached. Besides, values fluctuate more strongly between 0.8 and 0.9, which is presumably related to a declining test sample size.

The performance of the model predicting innovation expenditures constantly increases until a training sample size of about 0.85. It has a comparatively large drop afterwards and tends to be more volatile in general. Both can be explained by a much smaller overall sample size for this indicator. For instance, a train/test split of 0.5 implies fewer absolute observations included in the training sample. Also, the test sample is always smaller, which makes the evaluation of the performance less robust. Furthermore, the increasing trend indicates that the model will continue to improve if we would add more observations. AUC values based on training sample sizes between 0.75 and below 0.85 percent fluctuate around the AUC value pointed out in Table 5.

In summary, it can be stated that the analyzed website characteristics show a better performance in the prediction of product innovators and firms with innovation expenditures than of process innovators. Moreover, text-based features show a greater relative relevance.

To compare the relevance of single features across feature sets, the ten most important predictor variables measured by the MDI are displayed in Fig 3 for each 'all' feature model, respectively.

Three features exist that nearly always appear among the 10 most relevant: The total number of characters (*text_length*), the number of subpages (*nr_subpages*) (this feature only appears on the twelfth position for process innovators), and the share of English language (*english_language*). A further investigation of the top 100 most relevant features (see S1 Table) reveals that additional website characteristics exist with some general relevance. The words 'worldwide', 'innovative', 'application', 'to develop', 'product', 'technology' (all translated), the word 'system' as well as certain LDA topics, and the topic popularity index (*pop_score*), incoming (*incoming_links*), outgoing (*outgoing_links*) as well as social media hyperlinks (*social_media*), the Flesch-reading-ease score (*flesch_score*), the loading time of a website (*load_time*), and the share of numbers (*share_numbers*) are among the 100 most relevant features for every indicator. This shows that particular website characteristics exist, which have some relevance across indicators. In contrast, it is also noteworthy that features exist that show a large difference in the descriptive statistics but seem less important when predicting the innovation status. For example, the emerging technology term dummy never appears among the top 10 features for any indicator and is also not frequently observed among the top 100 features. Furthermore, some features are more relevant for certain innovation indicators than for others. For instance, IT-related features seem to be highly relevant for product innovators. The IT-related LDA topics 35 ("ICT infrastructure") and 134 ("business software") as well as the words software and system are (only) among the top 10 features for this indicator. Besides, LDA topic 7 with keywords linked to product experience and the word 'application' appear among the top 15 features.

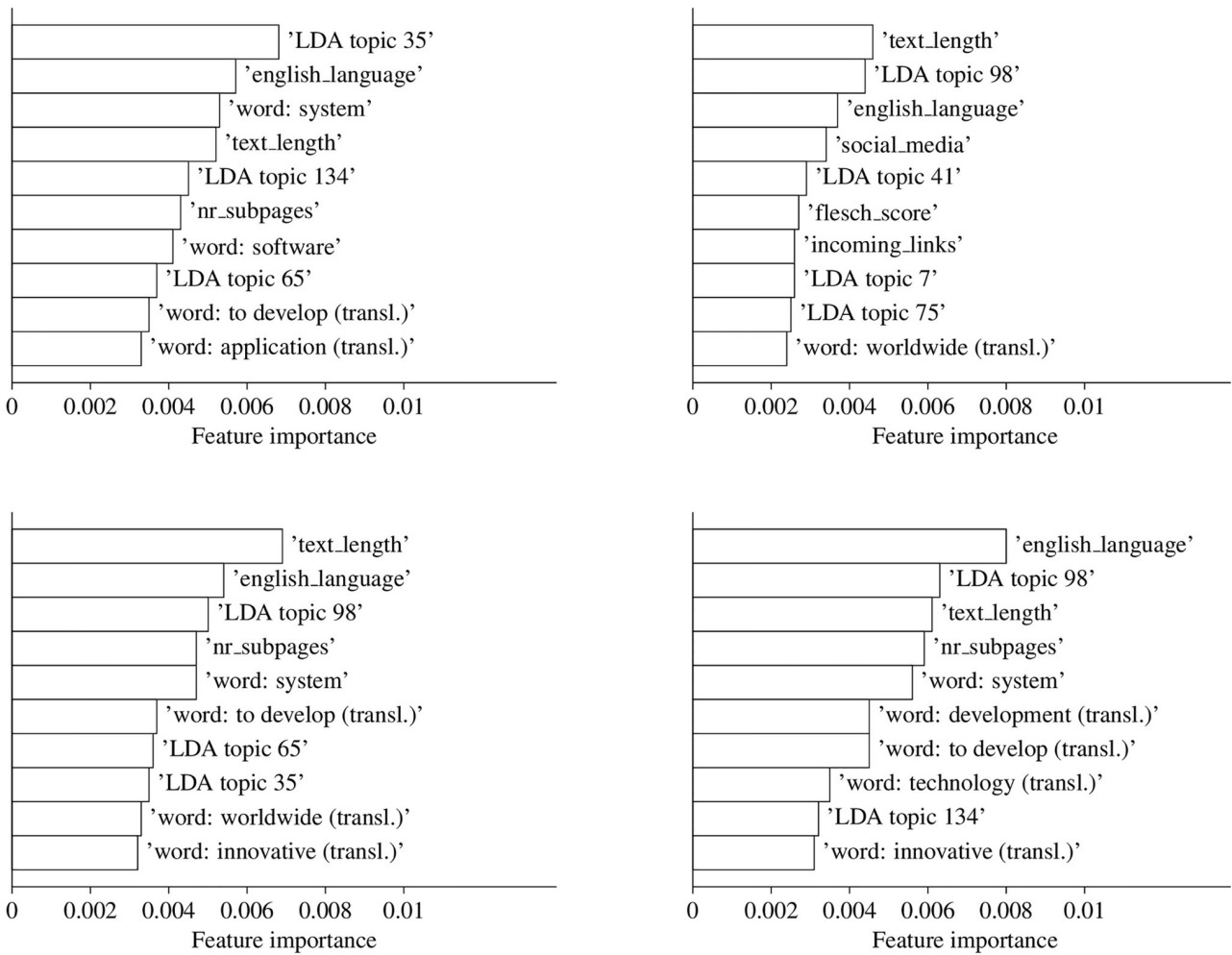

**Fig 3. Feature importance values for 'all' feature models.** For instance, a value that is two times larger implies that the mean decrease in impurity of the related feature is twice as high. Product innovators (top left), process innovators (top right), innovators (bottom left) and firms with innovation expenditures (bottom right) as target variable.

On the contrary, the research & development related LDA topic 98 is more important when estimating process innovators and firms with innovation expenditure. Besides, the LDA topic 65 occurs in Fig 3 for product innovators and innovators, which should be related to a negative relationship to innovation activity, as the descriptive statistics show that this LDA topic is more likely to appear on websites of firms with no innovation activity. With respect to process innovators, it should be mentioned that only a single word can be found in the 10 most important features and it is the only indicator that has 'network' features among its top 10. Furthermore, it is also interesting that the bottom left part of Fig 3, which relates to innovators, is at least for most features a combination of the most relevant features for product and process innovators. Last but not least, research & development related words are highly important for predicting firms with innovation expenditures.

## 7 Discussion

Descriptive statistics as well as our fitted Random Forest models show that website characteristics are relevant predictors for firm-level innovation activity. We see a significant difference in

means between innovative and non-innovative firms for most of our features. For each innovation indicator, Random Forest models using all features jointly show almost always a considerably higher performance than the baseline prediction with respect to the presented evaluation metrics. Moreover, the McNemar test confirms a significant difference to baseline predictions for all models. Also, our results are in line with [30]. Their statistical model has reached a similar accuracy for product innovators only observed in one MIP wave.

Our exercise also reveals—especially when predicting product innovators and firms with innovation expenditures—that 'text' features are relatively more important than 'meta' and 'network' features. Besides, we see a pattern regarding the most important characteristics independent of different target variables: Across indicators, the total number of characters, the number of subpages and the share of English language belong always to the most relevant. It is also noteworthy that these features are more important than the word "innovative". This finding suggests that website size and language should be considered for different types of website-based innovation indicators, which has not been done in previous studies. Meeting expectations, features that show insignificant differences in Table 3 almost never belong to the top 10 most relevant features in Fig 3. An exception is the *flesch_score* in the case of process innovators. Furthermore, considering the poor performance of the 'meta' feature models and the result that 'text' is the most relevant feature set, the relevance of website size is quite counterintuitive. One has to consider, however, that the importance of features is considered separately. The relevance of, e.g., the number of subpages is compared to the relevance of single words. If all words appearing in the term-document matrix would be considered jointly instead, their aggregated relative relevance would lie between 74 and 77 percent, depending on the indicator. This perspective illustrates why 'text' features and in particular textual content are still much more important for an accurate prediction. Nonetheless, as explained before, relative MDI importance should always be considered cautiously as it is affected by multicollinearity. Other web-based features may exist that possess predictive power and have not been considered in our analysis. These features would most likely change the result. Furthermore, it would also impact relative MDI importance, if this study's website data would be complemented with information from other sources, for example, non-web data from the MUP. In this case, innovation activity could potentially be predicted more accurately. However, we have deliberately decided against adding non-web data to our analysis, since this study focuses on the comparison of website information, which is up-to-date and freely accessible for everyone. Nonetheless, it would certainly be interesting to investigate in a further study the effect of adding additional non-web data. For potentially relevant features see [31].

Another aspect that we want to emphasize is the fact that features which are highly important for one indicator usually relate to its form of innovation activity. We see this as a strong indication that models use relevant information. Especially for firms with innovation expenditures, the selected word-based features appear particularly convincing. Terms like "to develop" (transl.) and "technology" (transl.) are highly ranked and have a very strong and direct connection to research & development expenditures. Another example is that the product experience related LDA topic 7 (top 15 most important features) and the term 'application' have a high importance for product innovators. Additionally, the 10 most relevant features of product innovators have a clear focus on information and communication (ICT) technologies, which is in line with the innovation spawning characteristic of ICT as well as the result of [49]. They find that ICT investment intensity is positively associated with innovation and stronger linked to product than to process innovation. Moreover, firms have a great incentive to present new products on their websites, process innovators, however, have a smaller incentive to announce innovation activity because new processes are less relevant for most website visitors. This might explain why results show a better predictive performance for product innovators than

for process innovators and for innovators in general. In addition, only a single word appears among the 10 most relevant features of process innovators and, even though this model differs on a higher significance level, 'text' features alone do only lead to slightly better predictions than 'meta' and 'network' features. This result supports the assumption that process innovations are often not mentioned explicitly.

Regarding innovators, most of its top 10 features either appear in the product or process innovator ranking and the predictive performance of the 'all' feature model lies between both as well. This result meets our expectations as the innovator target variable is a combination of product and process innovators.

Interesting is also the fact that, contrary to our expectations, some features are not that relevant. For instance, even though the descriptive statistics show a large difference between innovative and non-innovative firms, the emerging technology dummy does not seem to be very decisive for predictions. Looking at the Pearson correlation coefficients between this and all other features reveals that the emerging technology dummy has a comparatively strong relationship with other features. Hence, their relative MDI importance is probably ranked lower due to multicollinearity. Besides, even though the descriptive statistics do not show a significant difference for every form of innovation activity, the Flesch-reading-ease score, the loading time of a website, and the share of numbers appear to be relevant for every indicator (according to the 100 most relevant features). These features, however, do not relate strongly to other features and might, therefore, provide some extra information. Hence, they are relatively relevant despite small differences.

Although we show a clear link between website characteristics and innovation status, the predictive performance of our models leaves room for improvement as we, for example, still misclassify the existence of innovation expenditures for a considerable share of firms. Predictions might perform slightly better if neural networks were used. Our main criteria for choosing a Random Forest approach are the explainability of results and the fact that nonlinear relationships can be learned. Neural networks unfortunately do not offer a direct possibility to disclose decision processes. Hence, there is a trade-off, which often occurs in practice, between performance and explainability. If explainability is not necessary, predictive performance can most likely be improved by neural networks. Within our sample, there can be of course also innovative firms that do not mention their innovation activity (implicitly or explicitly) on their website. In other words, some inaccuracy might relate to the nature of our data. In particular, product innovators, process innovators, and innovators might suffer from noise as they cover a three year span. Websites can change a lot during this period. Comparatively good results for firms with innovation expenditures could be explained by the fact that this data is observed on an annual basis. Solving this matching problem seems to us a necessary step to improve predictions. Nonetheless, text data is always noisy and models with perfect accuracy are almost never identified.

Furthermore, it could be criticized that website-based innovation indicators can only be applied to firms that have a website. Another point of criticism would be that it could cause noise if for marketing purposes firms falsely claim on their website that they are innovative. The MIP contains self-reported data as well, however, firms do not have the incentive to make false declarations as answers should not affect their public image. For this reason, we expect MIP data to reveal the actual innovation status and we consider the usage of MIP-based information as target variables as a solution to the problem of false declarations of innovation activity on firm websites. Besides, patent data could have also been used as an alternative target variable. However, patent-based indicators suffer from large time lags and rather measure inventions than innovations.

## 8 Conclusion

In this research article, we contribute to the discussion on whether web-based innovation indicators are a feasible alternative to survey-based innovation indicators. We conduct our analysis with data on 4,487 German firms which reported different forms of innovation activity in a large-scale questionnaire-based survey (the MIP 2019). We extract website texts, additional website-related meta information as well as hyperlinks of these firms and use the information to predict firm-level innovation activity reported in the MIP. The performance of our machine learning models shows that website characteristics unambiguously relate to MIP-based innovation indicators. Furthermore, we find that website characteristics better predict product innovators and firms with innovation expenditures than process innovators. Hence, website characteristics rather appear to be suitable for measuring only certain aspects of innovation. Additionally, the importance of certain website characteristics varies between indicators. Accordingly, different features should be taken into account depending on the kind of innovation activity that is analyzed. Lastly, our work and related studies show that state of the art web-based predictive modeling cannot fully replace traditional surveys as error rates remain quite high. However, our models provide information about innovation activities that can be quickly updated, are on a very granular level (firm-level), and are less expensive than questionnaire-based surveys.

## Supporting information

**S1 Fig. Firm size distribution.** Firm size distribution for the estimation and the full MIP 2019 sample.
(TIF)

**S2 Fig. Firm sectors.** Firm distribution by sector for the estimation sample and the full MIP 2019 data set.
(TIF)

**S3 Fig. AUC values for different splits between training and test sample.** Line plot that illustrates for each indicator how AUC values of the 'all' feature model increase if the train/test split changes from (0.1/0.9) to (0.9/0.1) in steps of 0.01.
(TIF)

**S1 Table. Most relevant features.** Table with top 100 most relevant features for product innovators, process innovators, innovators and innovation expenditure.
(PDF)

**S2 Table. Software packages.** Details on used software packages.
(PDF)

**S3 Table. Details on learned hyperparameters.** Details on hyperparameter combinations of fitted models.
(PDF)

**S1 Text. Technological terms.** Full list of utilized technological terms in German and English language.
(PDF)

**S2 Text. Feature generation.** Detailed information on the feature generation.
(PDF)

## Acknowledgments

We thank Irene Bertschek, Reinhold Kesler, Christian Rammer, Bettina Schuck, Dominik Rehse, Thomas Niebel and Tobias Gließner for valuable inputs.

## Author Contributions

**Conceptualization:** Janna Axenbeck, Patrick Breithaupt.

**Data curation:** Janna Axenbeck, Patrick Breithaupt.

**Formal analysis:** Janna Axenbeck, Patrick Breithaupt.

**Methodology:** Janna Axenbeck, Patrick Breithaupt.

**Software:** Janna Axenbeck, Patrick Breithaupt.

**Validation:** Janna Axenbeck, Patrick Breithaupt.

**Visualization:** Janna Axenbeck, Patrick Breithaupt.

**Writing – original draft:** Janna Axenbeck, Patrick Breithaupt.

**Writing – review & editing:** Janna Axenbeck, Patrick Breithaupt.

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
