## [Decision Letter · Decision Letter 0]

9 Dec 2020

PONE-D-20-34158

Innovation indicators based on firm websites – Which website characteristics predict firm-level innovation activity?

PLOS ONE

Dear Dr. Axenbeck,

Thank you for submitting your manuscript to PLOS ONE. After careful consideration, we feel that it has merit but does not fully meet PLOS ONE’s publication criteria as it currently stands. Therefore, we invite you to submit a revised version of the manuscript that addresses the points raised during the review process.

We look forward to receiving your revised manuscript.

Kind regards,

Khanh N.Q. Le

Academic Editor

PLOS ONE

Journal Requirements:

2. We noted in your submission details that a portion of your manuscript may have been presented or published elsewhere.

"A previous version of this work ``Web-based innovation indicators - Which firm website characteristics relate to firm-level innovation activity?" is circulating as a discussion paper and can be accessed through http://dx.doi.org/10.2139/ssrn.3542199. A copy of this version is uploaded."

Please clarify whether this publication was peer-reviewed and formally published. If this work was previously peer-reviewed and published, in the cover letter please provide the reason that this work does not constitute dual publication and should be included in the current manuscript.

Reviewers' comments:

Reviewer's Responses to Questions

**Comments to the Author**

1. Is the manuscript technically sound, and do the data support the conclusions?

Reviewer #1: Partly

Reviewer #2: Yes

2. Has the statistical analysis been performed appropriately and rigorously? 

Reviewer #1: N/A

Reviewer #2: Yes

3. Have the authors made all data underlying the findings in their manuscript fully available?

Reviewer #1: No

Reviewer #2: Yes

4. Is the manuscript presented in an intelligible fashion and written in standard English?

Reviewer #1: Yes

Reviewer #2: Yes

5. Review Comments to the Author

Reviewer #1: In this article, the authors have performed Firm-level innovation activity based on the questionnaires. after addressing the following, this article may be recommended for publication.

-- The content in the figures is not clear.

-- There are several approaches in the literature are using the machine learning based algorithms to achieve the same objective. Why the authors not consider machine learning based algorithms to address the challenge?

-- The comparison results needed improvement and the presented results are not sufficient to judge the performance of the proposed work

-- The conclusion of this article is very lengthy. Recommended to provide the summary in less than 250 words (recommended conclusion length for many publications).

-- Provide the references of the data sets used in this article, from where authors consider the data set.

-- Perform the statistical analysis on the results to justify the accuracy of the results achieved.

Reviewer #2: A well written manuscript. I will suggest the authors to provide with more details about the classification methods used, criteria of classification and cut off. The authors should add the validation method used for the different models developed.

6. PLOS authors have the option to publish the peer review history of their article (what does this mean?). If published, this will include your full peer review and any attached files.

Reviewer #1: No

Reviewer #2: No

---

## [Author Response · Author response to Decision Letter 0]

11 Mar 2021

Dear Mr. Le, dear Reviewers,

We would like to extend our sincere gratitude and appreciation to you for reviewing our resubmission of the paper “Innovation indicators based on firm websites – Which website characteristics predict firm-level innovation activity?”. We understand the points raised in the first review and are convinced that your suggestions are of great value for our work. We have taken them all very seriously and were able to implement them. For ease of reference, we have numbered your comments and our responses below. The corresponding changes in the manuscript are highlighted in blue. When we make reference to certain lines of our manuscript, we refer to the highlighted manuscript.

##################

Comment 1 (raised by editor)

Please ensure that your manuscript meets PLOS ONE’s style requirements, including those for file naming.[...]

Response 1

Thank you for this comment. We paid close attention to PLOS ONE’s style requirements and adjusted our manuscript accordingly. We indicated corresponding changes as stated above. Furthermore, we adjusted file names when necessary.

Comment 2 (raised by editor)

We noted in your submission details that a portion of your manuscript may have been presented or published elsewhere. [...] Please clarify whether this publication was peer-reviewed and formally published. If this work was previously peer-reviewed and published, in the cover letter please provide the reason that this work does not constitute dual publication and should be included in the current manuscript.

Response 2

We want to apologize that we did not clarify this in our first submission. A previous version of our manuscript was published as “ZEW discussion paper”, which (very similar to preprints) forego formal peer review and publication. This enables researchers to circulate results timely and to receive early feedback. Hence, none of the work included in our manuscript was previously peer-reviewed or formally published. For clarification, we updated our cover letter in this regard.

Comment 3 (raised by editor) 

We note that you have indicated that data from this study are available upon request. PLOS only allows data to be available upon request if there are legal or ethical restrictions on sharing data publicly. [...] In your revised cover letter, please address the following prompts:

b) If there are no restrictions, please upload the minimal anonymized data set necessary to replicate your study findings as either Supporting Information files or to a stable, public repository and provide us with the relevant URLs, DOIs, or accession numbers

Response 3

Much to our deep regret, we cannot make the employed data publicly available. The data sets for which we indicated that they are available upon request, the Mannheim Innovation Panel (MIP) as well as the Mannheim Enterprise Panel (MUP), entail highly confidential information on firm turnover, product margins and personal information about survey respondents. Therefore, the data cannot be made available to the public as this is imposed by the EU-GDPR as well as by the German Data Protection Act (BDSG). However, for scientific purposes it is possible to request access at research data centers. Contact details of the ZEW Research Data Centre (ZEW-FDZ), which is the research data center that administers MIP and MUP data, can be retrieved from https://www.zew.de/en/research-at-zew/zew-research-data-centre-zew-fdz. The website data can also not be published as this is imposed by the German Copyright Act (§60d III p. 2). The data is stored in ZEW’s Dark Archive for the purpose of adherence to scientific standards and proving correct scientific work. However, for scientific review and verification it is possible to request access at https://www.zew.de/en/das-zew/serviceeinheiten/zentrale-dienstleistungen/bibliothek/. As requested, we now address this issue in our revised cover letter and provide contact information for data access inquiries.

Comment 4 (raised by Reviewer 1)

The content in the figures is not clear.

Response 4

We want to apologize that the content of our figures was not clear in our first submission. In order to increase the clarity of our figures, we made the following changes. Firstly, we renamed axis labels as well as elements of figure legends. Secondly, we added a sentence to each figure’s caption that exemplifies how to interpret data points of the respective image. We sincerely hope that the content of each figure is now much more understandable.

Comment 5 (raised by Reviewer 1)

There are several approaches in the literature are using the machine learning based algorithms to achieve the same objective. Why the authors not consider machine learning based algorithms to address the challenge?

Response 5

Thank you for this question, we gave a lot of thought to what machine learning algorithm suits best and are pleased to answer this question. By “machine learning” algorithm you are probably referring to neural networks, which are presumably the most prominent group of machine learning algorithms. We chose the Random Forest algorithm, which is also a machine learning algorithm, e.g., [1] and [2], over a neural network for the following reason. Neural networks possess comparatively high predictive power, but have the disadvantage that they do not reveal the relative importance of each variable. However, comparing single features with respect to how important they are for the prediction of each firm’s innovation status (explainability) is a crucial part of our study (e.g., see lines 616 – 655). The Random Forest algorithm has the advantage that the feature importance can be directly calculated, for instance, by means of the relative MDI, which is explained in the method section (lines 351 – 372). Previous empirical studies show that neural networks only perform marginally better than predictions based on the Random Forest algorithm, e.g., [3]. Hence, the Random Forest algorithm allows us to calculate feature relevance, while having a predictive performance comparable to neural networks. In our manuscript, we already discuss the trade-off between predictive performance and explainability (lines 746 – 752). Furthermore, it is not uncommon to apply the Random Forest algorithm for prediction tasks like ours. For instance, [4], a study also published in PLOS ONE, uses this algorithm in a similar context. To emphasize more strongly that the Random Forest algorithm is considered as machine learning, we added [2] and [5] to our reference list. Moreover, we included a short paragraph in our method section briefly describing the Random Forest algorithm and why it is considered as machine learning (lines 341 – 350).

Comment 6 (raised by Reviewer 1)

The comparison results needed improvement and the presented results are not sufficient to judge the performance of the proposed work

Response 6

We took this comment very seriously. 

To improve the overall validity of our model, we decided to modify our cross-validated grid-search and changed the metric that is applied to optimize hyperparameters and to select models. In our first submission, accuracy was used, which is now replaced by the AUC value. We prefer the AUC value as it is independent of a classification threshold (cut-off value). This modification changes results marginally. For instance, the AUC value of the ‘all’ feature model predicting innovation expenditure is now slightly improved, but the accuracy is a little lower. We updated altered results as well as the discussion in our manuscript accordingly and tracked changes (lines 498 – 772).

To enable readers to better judge the performance of the presented work, we conducted the following changes in our manuscript. As further evaluation metrics, we added precision and recall for both classes (innovative and non-innovative firms) to Table 5. Both evaluation metrics allow us to make judgements on the predictive performance for a single class. Hence, they allow for more detailed conclusions on how accurate predictions are only for innovative firms as well as only for non-innovative firms. In line with the extension of presented evaluation metrics, we now describe them more explicitly in our method section (lines 389 – 420). For instance, we added equations referring to the calculation of precision, recall, accuracy and F1-score. Furthermore, we added details on the cross-validation procedure (lines 435 – 442) as well as on how we measure feature importance (lines 351 – 372).

To sufficiently judge the robustness of the presented results, we conducted the following additional analysis. For each indicator, we fitted Random Forest models using all features and different split sizes between our training and test sample. We fitted models using all possible combinations of train/test splits between 0.1/0.9 and 0.9/0.1 (in steps of 0.01). The corresponding change of respective AUC values on the test sample for each indicator with respect to an increasing training sample is displayed in S3 Fig “AUC values for different splits between training and test sample”. In general, we find that AUC values reported in Table 5 are robust to different train/test splits if the number of observations in the training sample becomes sufficiently large (lines 593 – 610).

Furthermore, to compare fitted models and to measure whether they significantly differ from baseline estimates we applied a McNemar test, which is explained in more detail in Response 9.

Comment 7 (raised by Reviewer 1)

The conclusion of this article is very lengthy. Recommended to provide the summary in less than 250 words (recommended conclusion length for many publications).

Response 7

Thank you for raising this point. We condensed our conclusion to the recommended length.

Comment 8 (raised by Reviewer 1)

Provide the references of the data sets used in this article, from where authors consider the data set.

Response 8

We apologize for not providing a detailed reference with respect to the employed data sets. For the Mannheim Innovation Panel (MIP), we now include a reference in our manuscript with a DOI referring to the institution from where we retrieve the data [6] (line 155). Unfortunately, it exists no DOI for the Mannheim Enterprise Panel (MUP). However we are able to refer to an article which describes the data thoroughly [7] (line 167). Furthermore, in our cover letter we now provide contact information of the research data center that administers MIP and MUP data (https://www.zew.de/en/ research-at-zew/zew-research-data-centre-zew-fdz). We collected most website information by using the ARGUS web-scraper, which is described in detail in the following article [8]. In our first submission, we already cited this paper, but we did not re-cite it when we were referring to the ARGUS web-scraper. We now also refer to this article at this point. Additionally, we included the GitHub page from where the ARGUS web-scraper can be downloaded as a further source in our manuscript [9] (lines 174 –176). Unfortunately, we are not allowed to publish the scraped websites due to the German Copyright Act (§60d III p. 2), but data is stored in ZEW’s Dark Archive and it is possible to request access at https://www.zew.de/en/ das-zew/serviceeinheiten/zentrale-dienstleistungen/bibliothek/. Respective contact details are also now provided in our revised cover letter. We also refer to this issue in Response 3. Additionally, we provide very detailed supporting information on the calculation of our web-based features in S2 Text “Feature generation”.

Comment 9 (raised by Reviewer 1)

Perform the statistical analysis on the results to justify the accuracy of the results achieved.

Response 9

Thank you for making this comment. In our opinion, addressing this point highly improved the quality of our results. In order to justify the accuracy of results achieved, we now analyze whether fitted models significantly differ from baseline predictions by means of a McNemar test [10]. To be more precise, the McNemar test measures in our context whether predictions from two machine learning models significantly disagree with each other. We chose the McNemar test because it is in comparison to a t-test less prone to type I errors [11]. We adjusted our manuscript accordingly. In our method section, we now describe the McNemar test, present the equation to calculate the test statistic, and explain how to interpret the test statistic with respect to our overall research question (lines 377 – 388). Furthermore, the McNemar test confirms that the divergence between fitted model and baseline prediction is highly significant for most combinations of feature sets and indicators. We describe respective test outcomes in our results section and also relate them to the presented evaluation metrics (lines 571 – 585).

Comment 10 (raised by Reviewer 2)

A well written manuscript. I will suggest the authors to provide with more details about the classification methods used, criteria of classification and cut off.

Response 10

We want to thank you very much for the appreciation and your helpful comments. In response to your request, we added the following clarifying information to our manuscript. With respect to our classification method, we now provide more details on the Random Forest algorithm in our manuscript. For instance, we now explain its intuition and why we selected this particular machine learning algorithm (lines 341 – 350). We also added a further reference that describes the Random Forests algorithm formally [2].

A Random Forest consists of single decision trees. Within each tree the data is split at nodes, generating “purer” branches. We use the “decrease in impurity” as split criterion for branches, which we now explain extensively in lines (lines 351 – 366). To improve out-of-sample predictions, we combine 5-fold cross-validation with a grid-search. The metric for model selection of the cross-validated grid-search is the AUC (“Area Under the Curve”) (lines 429 – 442). Our Random Forest model is learning the relationship between innovation activity and web-based features, which are thoroughly described in our data section as well as in S2 Text “Feature Generation”. Classification thresholds of each feature vary from tree to tree, but the average importance of one feature in the entire forest can be calculated by means of the “mean decrease in impurity”, which is also now explained in detail in lines 364 – 372. Furthermore, we present the ten most important features for each indicator in Fig 3 “Feature importance values for ‘all’ feature models” and describe them in our results section (lines 616 – 655). By applying the grid-search, we optimize hyperparameter combinations with regard to ‘number of trees’, ‘maximum tree depth’ and ‘minimum impurity decrease’. For all other hyperparameters we use default values provided by scikit-learn as now clarified in lines 432 – 433. Additionally, we now point out patterns of optimized hyperparameters (lines 586 – 592). In the interest of profound documentation, we now provide a detailed overview of learned hyperparameter combinations for each model in S3 Table “Details on learned hyperparameters”.

We use 0.5 as cut-off value in all fitted models, as this value is most commonly used and we do not prefer one class over the other (lines 406 – 411). Motivated by our request, we also experimented with a classification thresholds based on the Youden’s J statistic [12]. However, estimating the optimal cut-off value would require additional training data and we figured out that our sample size is not large enough to generate two different training samples.

Comment 11 (raised by Reviewer 2) 

The authors should add the validation method used for the different models developed.

Response 11

First, we want to apologize that the validation method was not transparent in our initial submission. We believe that your comment has helped us to improve the scientific rigor of the paper.

For validation, we use a test sample, which is indeed a “hold-out” sample that is never used for training. We now emphasize this more strongly in lines 424–427. Furthermore, the presented results in Table 5 are also calculated on the test sample. We noted that we did not point this out explicitly in our first submission and the information is now added to the title of Table 5. As previously stated, we apply a grid-search with a 5-fold cross-validation on the training sample to tune hyperparameters for all of our models. In our revised manuscript, we now describe the k-fold cross-validation procedure in much more detail (lines 434–442). We have also revised evaluation metric descriptions and now present them more formally (lines 389–420).

##########

Additionally, we conducted the following changes. Patrick Breithaupt is also now a member of the Faculty of Economics at Justus-Liebig-University Giessen and we accordingly added this information to its affiliations. We changed one sentence in our abstract and added another one due to altered results. Another sentence was changed in lines 317 – 318. We added information about the selected random seed (lines 443 – 445). We moved one paragraph from the method section to the results section and slightly changed the wording. The paragraph is now placed at lines 501 – 504. We also improved the language in the entire results section. To further improve clarity, we added variable names when we describe results with respect to feature importances (lines 619 – 630). We replaced one sentence in the discussion (lines 711 – 714) and deleted two (lines 720 – 722 and lines 726 – 728). For the ease of reading, we just reduced our argument to one example at lines 633 – 636 and at lines 730 – 735.

Moreover, we are now more explicit in line 769.

We sincerely hope that we have been able to address your comments in accordance with your expectations.

Kind regards,

Janna Axenbeck & Patrick Breithaupt

References

1. Zhang C, Ma Y. Ensemble Machine Learning. Boston MA; Springer. 2012.

2. Breiman L. Random Forests. Machine Learning. 2001; 45(1): 5–32.

3. Ahmad MW, Mourshed M, Rezgui Y. Trees vs neurons: Comparison between random forest and ANN for high-resolution prediction of building energy consumption. Energy and Buildings. 2017; 147: 77–89.

4. Gandin I, Cozza C. Can we predict firms’ innovativeness? The identification of innovation performers in an Italian region through a supervised learning approach. PLOS ONE. 2019 June; 14(6): e0218175.

5. Mohri M, Rostamizadeh A, Talwalkar A. Foundations of Machine Learning. Cambridge MA; MIT press. 2018.

6. Rammer C, Peters B, Doherr T, ZEW – Leibniz Centre for European Economic Research (ZEW). Mannheim Innovation Panel (MIP; data type 1); 2019. ZEW-FDZ https://doi.org/10.7806/zew.mip.2019.V1.suf

7. Bersch J, Gottschalk S, Müller B, Niefert M. The Mannheim Enterprise Panel (MUP) and firm statistics for Germany. ZEW Discussion Paper. 2014; (14-104). Available from: http://ftp.zew.de/pub/zew-docs/dp/dp14104.pdf

8. Kinne J, Axenbeck J. Web mining of firm websites: A framework for web scraping and a pilot study for Germany. Scientometrics. 2020: 1–31.

9. Kinne J. ARGUS - An Automated Robot for Generic Universal Scraping; 2018. Available from: https://github.com/datawizard1337/ARGUS

10. McNemar Q. Note on the sampling error of the difference between correlated proportions or percentages. Psychometrika. 1947; 12(2): 153–157.

11. Dietterich TG. Approximate Statistical Tests for Comparing Supervised Classification Learning Algorithms. Neural Computation. 1998; 10(7): 1895-1923.

12. Youden WJ. Index for rating diagnostic tests. Cancer. 1950; 3: 32–35.

---

## [Decision Letter · Decision Letter 1]

22 Mar 2021

Innovation indicators based on firm websites – Which website characteristics predict firm-level innovation activity?

PONE-D-20-34158R1

Dear Dr. Axenbeck,

We’re pleased to inform you that your manuscript has been judged scientifically suitable for publication and will be formally accepted for publication once it meets all outstanding technical requirements.

Kind regards,

Khanh N.Q. Le

Academic Editor

PLOS ONE

Additional Editor Comments (optional):

Reviewers' comments:

Reviewer's Responses to Questions

**Comments to the Author**

1. If the authors have adequately addressed your comments raised in a previous round of review and you feel that this manuscript is now acceptable for publication, you may indicate that here to bypass the “Comments to the Author” section, enter your conflict of interest statement in the “Confidential to Editor” section, and submit your "Accept" recommendation.

Reviewer #1: All comments have been addressed

2. Is the manuscript technically sound, and do the data support the conclusions?

Reviewer #1: Yes

3. Has the statistical analysis been performed appropriately and rigorously? 

Reviewer #1: Yes

4. Have the authors made all data underlying the findings in their manuscript fully available?

Reviewer #1: Yes

5. Is the manuscript presented in an intelligible fashion and written in standard English?

Reviewer #1: Yes

6. Review Comments to the Author

Reviewer #1: The authors addressed all the comments and recommending this article for publication... Congratulations to the authors..

7. PLOS authors have the option to publish the peer review history of their article (what does this mean?). If published, this will include your full peer review and any attached files.

Reviewer #1: No

---

## [Editor Report · Acceptance letter]

25 Mar 2021

PONE-D-20-34158R1 

Innovation indicators based on firm websites – Which website characteristics predict firm-level innovation activity? 

Dear Dr. Axenbeck:

I'm pleased to inform you that your manuscript has been deemed suitable for publication in PLOS ONE. Congratulations! Your manuscript is now with our production department. 

Kind regards, 

on behalf of

Dr. Khanh N.Q. Le 

Academic Editor

PLOS ONE